# Characteristics and Discrimination of the Commercial Chinese Four Famous Vinegars Based on Flavor Compositions

**DOI:** 10.3390/foods12091865

**Published:** 2023-04-30

**Authors:** Yong Hu, Chuanyang Zheng, Haiyin Chen, Chao Wang, Xiyue Ren, Shiming Fu, Ning Xu, Panheng Li, Jinyi Song, Chao Wang

**Affiliations:** 1Key Laboratory of Fermentation Engineering (Ministry of Education), Hubei Key Laboratory of Industrial Microbiology, National “111” Center for Cellular Regulation and Molecular Pharmaceutics, Cooperative Innovation Center of Industrial Fermentation (Ministry of Education & Hubei Province), Hubei Research Center of Food Fermentation Engineering and Technology, Hubei University of Technology, Wuhan 430068, China; 2Suizhou February Wind Food Co., Ltd., Suizhou 431518, China; 3Zhongxiang Weicheng Fruit and Vegetable Professional Planting Cooperative, Jingmen 431999, China

**Keywords:** Chinese traditional vinegar, volatile, organic acid, amino acid, classification

## Abstract

Shanxi aged vinegar (SAV), Zhenjiang aromatic vinegar (ZAV), Sichuan bran vinegar (SBV), and Fujian monascus vinegar (FMV) are the representative Chinese traditional vinegars. However, the basic differential compositions between the four vinegars are unknown. In this study, compositions of commercial vinegar were investigated to evaluate the influence of diverse technologies on their distinct flavor. Unlike amino acids and organic acids which were mostly shared, only five volatiles were detected in all vinegars, whereas a dozen volatiles were common to each type of vinegar. The four vinegars could only be classified well with all compositions, and difference analysis suggested the most significant difference between FMV and SBV. However, SAV, ZAV, and SBV possessed similar volatile characteristics due to their common heating treatments. Further, the correlation of identification markers with vinegars stressed the contributions of the smoking process, raw materials, and *Monascus* inoculum to SAV, SBV, and FMV clustering, respectively. Therefore, regardless of the technology modification, this basic process supported the uniqueness of the vinegars. This study contributes to improving the standards of defining the characteristics of types of vinegar.

## 1. Introduction

Vinegar is manufactured worldwide using fermentable sugary materials. Unlike European vinegar, which is mainly produced from fruits, vinegars from grains are mainly manufactured in East Asia. In China, various traditional vinegars have been developed based on unique technology. Among them, the top representative vinegars comprise Shanxi aged vinegar (SAV), Zhenjiang aromatic vinegar (ZAV), Sichuan Bran vinegar (SBV), and Fujian monascus vinegar (FMV), which are known as the ‘four famous vinegars’ [1]. The popularity of the four vinegars is considerable in view of the fact that they comprise more than 13% of the market share in China (https://bg.qianzhan.com/report/detail/300/191211-4aa350d1.html accessed on 11 December 2019).

Traditional technologies usually adopt an open fermentation technique with natural inoculation. The biological process includes three stages: the saccharification of cereal starch with the use of a saccharifying agent, alcohol fermentation, where sugar–alcohol transformation is mainly functioned by *Saccharomyces cerevisiae*, and acetic acid fermentation (AAF) with acetic acid bacteria (AAB) [2] (Figure 1). After leaching and decoction, it is necessary to perform the aging process to enhance the flavor. In whole procedures, chemical and enzymatic reactions play key roles in the quality formation. For all vinegars, saccharification and alcohol fermentation could be maintained as part of the solid state. However, AAF could be divided into liquid-state fermentation (LSF), represented by FMV and solid-state fermentation (SSF), such as for SAV, ZAV, and SBV [3] (Figure 1). In FMV, water is added after alcohol fermentation, and liquor is filtered out and inoculated with liquid seeds for AAF. In the other three vinegars, the mixture of alcohol fermentation is mixed with a certain number of materials, including *pei*, bran, and chaff, generating only a small amount of free water throughout the process. Additionally, there are detailed technological properties unique to each of the ‘four famous vinegars’ (Figure 1). The prominences are as follows: layer-by-layer flipping vinegar *pei* is used in ZAV [2], much of the raw materials in SBV are made up of brans [4], a unique process of smoking vinegar *pei* is used in SAV [5], and FMV includes a special red koji (*Monascus*) and long-term fermentation [6]. Others include differences in the kinds and proportions of secondary materials, the control of fermentation, feeding strategies, and so on.

In industrial production, modified methods are applied to improve productivities and reduce production cost. Generally, the modifications are based on traditional methods [7]. While the preserved properties contributed to the similarity between traditional and industrial vinegar, the changed processes inevitably resulted in some different characteristics [7]. Xiao et al. [11] suggested that the volatile profiles could characterize commercial vinegars in terms of their types, fermentation method, and production area. However, another study has provided confused discrimination through making metabolomic comparisons [12]. In addition to industrial technologies, traditional processes are still used in some production methods. Regardless of the application of various technologies, it is undeniable that the ‘four famous vinegars’ have distinct flavor characteristics [13]. It is interesting to explore detailed chemical compositions and their relationship with the technologies.

In the present study, the profiles of the volatiles, free amino acids, and organic acids were investigated in commercially available ‘four famous vinegars’ to evaluate the influence of complicated industrial technologies on the characteristics of the vinegars. The results made clear that each type of vinegar has its own characteristics, and markers spanning volatiles, amino acids, and organic acids could be extracted to distinguish the four vinegars. The study supported the importance of the basic technology in maintaining the characteristics, which promoted the basis for defining the properties and developing quality standards for the ‘four famous vinegars’.

## 2. Materials and Methods

### 2.1. Chemicals and Samples

Eight commercial samples for each type of vinegar were randomly purchased from a local market. The product information was extracted from the labels, and this is outlined in Appendix A. Organic acid standards (formic acid, acetic acid, pyruvic acid, fumaric acid, oxalic acid, malic acid, pyroglutamic acid, tartaric acid, lactic acid, citric acid, and succinic acid), amino acid standards (Asp, Thr, Ser, Glu, Gly, Ala, Cys, Val, Met, Ile, Leu, Tyr, Phe, Lys, His, Arg, Pro, pyroglutamic acid, and GABA), butanal, rutin, gallic acid, catechin, *p*-coumaric acid, ferulic acid, caffeic acid, chlorogenic acid, and 2-octanol were obtained from Sigma-Aldrich (St. Louis, MO, USA). ABTS (2,2′-azino-bis(3-ethylbenzothiazoline-6-sulphonic), FRAP (ferric reducing/antioxidant power), DPPH (1,1-Diphenyl-2-picrylhydrazyl), K_2_S_2_O_8_, FeCl_3_, NaCl, Na_2_CO_3_, NaNO_2_, Al(NO_3_)_3_, NaOH, H_2_O_2,_ NH_3_.H_2_O, 2,4,6-tripyridyl-s-triazine, phosphate, soybean lecithin_,_ deoxyribose, trichloroacetic acid, fulin, methanol, and ethanol were purchased from Sinopharm Chemical Reagent Co., Ltd. (Shanghai, China).

### 2.2. Head Space Solid-Phase Micro-Extraction (HS-SPME)–Gas Chromatography–Mass Spectrometry (GC-MS) Analysis

Divinylbenzene/carboxen/polydimethylsiloxane (DVB/CAR/PDMS) 50/30 μm extraction fiber (Supelco, Bellefonte, PA, USA) was used to extract the volatiles. Agilent 7890A GC-MS (Palo Alto, CA, USA) was instrumented with an Agilent DB-WAX capillary column (30 m × 0.25 mm × 0.25 μm) (Palo Alto, CA, USA) and an Agilent 5975C mass selective detector (Palo Alto, CA, USA). The thermal desorption was executed at 250 ℃ for 4 min using the splitless mode. The GC oven temperature was maintained at 40 ℃ for 5 min, which then increased to 120 °C at a rate of 4 °C/min and was held for 25 min, until it increased to 200 °C at 8 °C/min and was held for 5 min, and it was then finally raised up to 260 °C at 8 °C/min and held for 40 min. The MS interface temperature, ion source temperature, and MS quadrupole temperature were set at 280 °C, 230 °C, and 150 °C, respectively. The ionization voltage was 70 eV. The mass ranges of MS were 33–450 amu with the scan rate of 0.7 scan/s.

Tentative identification was conducted according to mass spectra matching the NIST 08 Standard Reference Database and Wiley library using 2-octanol as the internal standard. Compounds were further identified by referring to the retention indices (RIs) of C_6_–C_33_ n-alkane series. Butanal was confirmed according to the MS of the reference standard since it fell outside of the RI range of C_6_–C_33_ n-alkane. Relative quantitation was obtained by measuring the relative peak area referring to 2-octanol (1 mg/L).

### 2.3. Free Amino Acid Analysis

The sample was added into the equal volume of sulfosalicylic acid (8%). The solution was centrifuged at 4 °C with 12,000 rmp for 15 min. The supernatant was diluted 150 times with hydrochloric acid (0.02 M). After filtration through a 0.22 μm syringe filter and ultrasonic vibration for 5 min, the supernatant was analyzed with an amino acid analyzer (L-8900, Hitachi, Tokoy, Japan) equipped with Hitachi dedicated ion exchange resin (4.6 mm × 60 mm × 3 μm) (Tokyo, Japan). The column temperature, pump flow rate, and pump pressure were set at 135 °C, 0.000–0.999 mL/min, and 0–20 kpa, respectively. Identification and quantification of amino acids were performed based on the standards retention time and calibration curves, respectively.

### 2.4. Organic Acid Analysis

Organic acid was determined using high-performance liquid chromatography (HPLC) (Waters 1515; Waters Co., Milford, MA, USA) equipped with an YMC-Pack ODS-AM C_18_ column (250 mm × 4.6 mm × 5 μm) (Tokyo, Japan). Briefly, 1 mL of sample was mixed with 9 mL of distilled water, and 1.92 mL of solution was further mixed with 40 μL of K_4_Fe(CN)_6_ solution (10.6%) and 40 μL of ZnSO_4_ solution (30%). Then, the supernatant was collected at 4 °C with 12,000 rpm for 15 min. The column temperature was held at 35 °C, and the wavelength of the UV detector was set at 210 nm. The mobile phase was 20 mM NaH_2_PO_4_ (pH 2.7) with a flow rate of 0.5 mL/min. Organic acids were identified according to the standards retention time, and quantification was carried out based on the calibration curves.

### 2.5. Antioxidant Activity (TAC) Analysis

For the ABTS method, 7 mM of ABTS solution and 2.45 mM of K_2_S_2_O_8_ solution were mixed to produce an ABTS radical cation (ABTS•^+^). The ABTS•^+^ solution was diluted with ethanol to an absorbance of 0.70 ± 0.02 at 734 nm. Then, 25 μL of sample or ethanol (control) was added into 2 mL of ABTS•^+^ solution and absorbance was measured at 734 nm. The ABTS antioxidant activity of the sample was relatively expressed as (A_ethanol_−A_sample_)/A_ethanol_ × 100% [14].

For DPPH, 25 μL of the sample was added to DPPH (2 mL, 62.5 μM) dissolved in methanol. After incubation for 30 min in the dark, the mixture was measured at 517 nm. The same volume of solvent displacing the sample was used to make the control.

For FRAP, the FRAP solution, which contains 25 mL of acetate buffer (300 mM, pH 3.6), 2.5 mL of 2,4,6-tripyridyl-s-triazine (10 mM), and 2.5 mL of FeCl_3_ (20 mM), was prepared. The sample (10 μL) was mixed with distilled water (1 mL) and was allowed to react with 1.8 mL of FRAP solution at 37 °C for 10 min and was then detected at 593 nm [14].

For TBARS, 5.0 g of soybean lecithin was sonicated in 500 mL of phosphate buffer (20 mM, pH 7.4) to obtain liposome. The reaction mixture, containing 50 μL of sample, 2 mL of liposome (1%), 0.1 mL of FeCl_3_ (25 mM), 0.1 mL of H_2_O_2_ (25 mM), 0.1 mL of ascorbate (25 mM), and 1.65 mL of phosphate buffer (20 mM, pH7.4), was incubated for 1 h at 37 °C. Then, 1 mL of butylated hydroxytoluene (20 mg/mL, dissolved in methanol), 1 mL of thiobarbituric acid (1%), and 1 mL of HCl (10%) were added. The mixture was boiled for 30 min and cooled until it reached room temperature. Following centrifugation at 3000× *g* for 20 min, the supernatant was detected at 532 nm [15].

For HRSA, the reaction mixture was prepared with 0.1 mL of sample, 0.5 mL of phosphate buffer (0.2 M, pH 7.4), 0.69 mL of deoxyribose (2.5 mM), 0.2 mL of FeCl_3_ (2 mM), 0.1 mL of ascorbate (1 mM), and 10 μL of H_2_O_2_ (0.1 M) and was incubated at 37 °C for 15 min. Then, 1 mL of trichloroacetic acid (2.8%) and 0.5 mL of thiobarbituric acid (1%) were added sequentially. After boiling for 10 min and cooling to room temperature, the mixture was prepared for measurement at 532 nm [14].

### 2.6. Determination of Total Polyphenol (TP) and Total Flavone (TF)

Gallic acid was used to prepare the calibration curve for the determination of TP. Briefly, 100 μL of the sample was diluted 10 times and mixed with 2 mL of fulin solution (10%) for 1 min. Then, 2 mL of Na_2_CO_3_ (7.5%) and 8 mL of distilled water were added. After shaking for 2 h in darkness, the absorbance was measured at 765 nm [16]. For TF, rutin was used to obtain the calibration curve. The sample (100 μL) was diluted 10 times and mixed with 0.5 mL of NaNO_2_ (10%). Then, 6 min later, 0.5 mL of Al(NO_3_)_3_ (10%) was added and the mixture was incubated for 6 min. Then, 4 mL of NaOH (4%) and 4.8 mL of distilled water were added. After 15 min, the absorbance was measured at 510 nm [16].

### 2.7. Determination of Phenolic Compositions

The sample was ultrasonically extracted with ethyl acetate in an ultrasonic cleaner (SB-5200 DT, Ningbo Scientz Biotechnology, Ningbo, China) at 200 W for 15 min. The extracts were filtered through a C_18_ solid-phase extraction column (Cleanert S, Agela Technologies Inc., Tianjin, China) and were eluted with a mixture of methanol (3 mL) and methanol containing 5% NH_3_.H_2_O (5 mL). The eluent was evaporated at 40 °C (EYELA Tokyo Rikakikai Co., Tokyo, Japan), dissolved in methanol, and filtered through a 0.45 µm membrane for HPLC detection (280 nm) (Agilent 1260, CA, USA), which was equipped with a YMC-Pack ODS-AM C_18_ column (250 mm × 4.6 mm × 5 μm) (YMC, Tokyo, Japan). A binary solvent system including solution I (99.9% water and 0.1% formic acid) and solution II (methanol) was used with a flow rate of 1.00 mL/min: 10 to 50% I (0–3 min), 50% I (3–16 min), 50 to 90% I (16–17 min), 90% I (17–19 min), 90 to 5% I (19–20 min), and 5% I (20–25 min). Compounds were identified referring to the retention time of the standards. Quantification was conducted using the standard curves.

### 2.8. Statistical Analysis

Measurements were repeated in triplicate. Statistics were presented as means of the average value ± standard deviation (SD). Significant differences were detected with a one-way analysis of variance (ANOVA) and Tukey’s test (SPSS 17.0) (SPSS Inc., Chicago, IL, USA). Orthogonal partial least squares discriminant analysis (OPLS-DA) can be used for feature extraction and the classification of components and can identify variables related to different classifications determining the contribution of variables to sample differences [17]. LDA effect size (LEfSe) can describe differences between groups and screen variables with the most significant differences between them [18]. Random forest is a machine learning algorithm based on decision trees which can perform feature selection from a large number of variables [19]. Partial least squares discrimination analysis (PLS-DA) is a multivariate statistical analysis based on multiple linear regression. It is also used to deal with the relationship between multiple independent or dependent variables [20]. OPLS-DA was performed with SIMCA 14.1. The differential components were imported into metaboanalyst 5.0 (https://www.metaboanalyst.ca accessed on 21 December 2022) for heat map analysis. Principal component analysis (PCA), PLS-DA, LEfSe, and random forest analysis was performed using R 4.0.2.

## 3. Results and Discussion

### 3.1. Comparison of the Volatiles

A total of 68 volatiles were identified from SAV (57), ZAV (46), FMV (45), and SBV (44) (Appendix A). In the samples, 20-30 volatiles were identified (Appendix A), which were similar to other studies of Chinese traditional vinegars [21,22], but were lower compared to balsamic vinegar [23,24]. The materials and whole technical process, including the aging process, should be the main factors which result in a difference. The combinational use of volatile extraction using an organic solvent would expand the overall profile of the volatiles [23].

The variety numbers and total contents showed no significant difference among four types of vinegar (Appendix A). There were 22 components (bold in Appendix A), of which 12, 14, 10, and 13 were shared by SAV, ZAV, FMV, and SBV, respectively. However, only five volatiles (ethyl acetate, acetic acid, furfural, hexanoic acid, and benzaldehyde) were shared by all vinegars, suggesting that technical characteristics contributed to the different aroma profiles of the four vinegars [25]. Acetic acid is the most important flavor of vinegar, and ethyl acetate could be commonly generated from microbial catalysis [26]. It is not surprising that the two compounds were shared by all samples. Both hexanoic acid and benzaldehyde might be produced mainly during alcohol fermentation [27] by *Kluyveromyces* [28,29] and *Rhizopus* [30], respectively. Furfural could be widely formed by the acid hydrolysis of pentosan [31]. The contents of all these five volatiles fluctuated among the four vinegars. Additionally, the contents of many volatiles changed greatly, even for samples from a type of vinegar, which resulted in larger SDs than the average values.

The identified compounds were grouped as esters (14), pyrazines (7), alcohols (5), acids (10), furans (4), phenols (8), aldehydes (10), ketones (4), and others (6). For each type of vinegar, there seemed to be more varieties of esters (3–8 kinds, 4.06–24.56%), acids (3–7 kinds, 12.17–28.43%), and aldehydes (3–7 kinds, 8.60–32.49%) compared to the other classes (Appendix A). In addition, phenols and ketones have more varieties in SAV and ZAV, respectively, and the number of aldehydes was significantly less in FMV. For each type of vinegar, the esters (1.15–65.61%) and acids (19.73–56.16%) appeared to have the most abundant contents (Appendix A). Additionally, furans were characteristically rich in ZAV and SBV (Appendix A). The abundance of acids and furans were mainly caused by acetic acid and furfural, respectively. When the two compounds were excluded, the esters dominated in vinegars with varieties between 35.13 and 40.38% and contents between 42.34 and 73.93%, confirming the key role esters play in the aroma characteristics [32]. The powerful esterification capacity of *Monascus* inoculum should make a higher content of esters in FMV (Appendix A) [33]. The prolonged fermentation of FMV further increased the contents through chemical esterification [26].

### 3.2. Heating Treatments Modified the Aroma Characteristics

Since any 1 of the 22 shared components was detected fully at least in a type of vinegar, they can stably represent the characteristics of vinegars. PCA with the 22 components separated FMV from the others (Figure 2A), which was similar to the results of the analysis of all 68 volatiles (Figure 2B). In the top 10 contributing compounds, 2-methylpropyl acetate could be a marker to distinguish LSF vinegar from SSF ones because of its existence in all FMVs but not in others (Appendix A). Six compounds (methyl-2-phenyl-2-hexena, 2,3-butanediol, 5-methyl-2-furancarboxaldehyde, furfural, dihydro-5-pentyl-2(3H)-furanone, and 2,3,5-trimethypyrazine), close to the groups of SSF vinegars (SAV, ZAV, and SBV), have been suggested to be related to the heating treatment [11,34,35,36]. The results implied the key role the heating process plays in modeling the characteristics of SAV, ZAV, and SBV [12]. To confirm this, TP, TF, and TAC were detected since phenolic compounds are mainly produced in thermal treatment [11]. Both TP and TF showed higher contents in SSF vinegars (Appendix A), and the TACs measured by five methods showed a liner correlation with either the TP or TF contents (Appendix A). Many phenolic compounds were detected in SAV (7) and ZAV (6) in previous studies [5,7] (Appendix A). Catechin was shared by all samples and made up the majority of the percentages in ZAV (46.05–48.32%) [7], FMV (93.53–99.3%), and SBV (72.10–73.33%). In addition to heat treatment, the raw materials, such as bran and chaff, which are used in SSF vinegars but not in FMV, may also contribute to the formation of flavor characteristics by thermal reaction [37]. For instance, the pentosane from chaff material contributed to the abundant furfural in SSF vinegars [38].

### 3.3. Comparison of the Amino Acids

The contents of total amino acids were richer in SAV and especially in SBV [1] (Figure 3A), which should be due to the utilization of a great deal of the proportion of protein-rich raw bran [39]. Supporting this, less bran was utilized in FMV and ZAV, and a low quantity of amino acids was detected.

Except Trp (not detected in all vinegars), His (not detected in one FMV), and Arg (not detected in two SAVs, three ZAVs, and all FMVs), other amino acids were all included in the samples (Appendix A). For each kind of vinegar, Ala (5.01–16.80% of total amino acids), Glu (4.31–17.42%), Phe (5.22–9.19%), Tyr (4.97–12.74%), and Val (2.65–9.46%) were the most abundant amino acids (Appendix A). In addition, FMV was rich in Ser (2.88–4.50%), and SBV contained rich Leu (7.77–9.51%), GABA (5.48–8.85%), and His (1.71–3.42%). The loss of Trp in vinegar has been attributed to decomposition when coexisting with amino acids, sugars, and aldehydes [40]. The transformation to off-flavors by lactic acid bacteria (LAB) might be another reason [41]. The absence of Arg in FMV was inconsistent with a previous observation [6], which might be due to the trace concentrations.

### 3.4. Comparison of the Organic Acids

Total amino acids and total organic acids were lower in ZAV and FMV (Figure 3B), and this resulted mainly from the lower acetic acid and lactic acid content. However, lactic acid and acetic acid were still the most abundant organic acids for all vinegars. Their contents together accounted for more than 88.86% of total organic acids (Appendix A). However, lactic acid was significantly higher in SBV (42.69–78.34%), which was consistent with a previous study [42].

In comparison, ZAV possessed a greater proportion of formic acid, citric acid, and oxalate acid, SBV possessed more malic acid and lactic acid, and FMV possessed more acetic acid and fumaric acid. Formic acid (0.06–0.82%), fumaric acid (0–0.01%), and malic acid (0–0.11%) had very low contents in all vinegars, which probably was due to the ubiquitous transformation through malolactic fermentation by LAB [43]. Several organic acids were not detected in the samples (Appendix A). Particularly, malic acid was absent in FMV, citric acid was absent in FMV and SBV, and pyroglutamic acid was absent in ZAV. All these acids have been reported in the vinegars [6,7,9]. The concentrations of these organic acids might be too low to be detected.

### 3.5. Clustering and Difference Analysis

Through PCA of amino acids, organic acids, or both of these, no type of vinegar could be separated, though the contents of total amino acids and total lactic acid were significantly higher in SBV (Figure 4A–C). However, when either all 68 volatiles or the 22 shared ones were used together with the amino acids and organic acids, the 4 types of vinegars could be classified well (Figure 4D,E), suggesting that a range of compounds represented the characteristics of the vinegars.

To verify the results above, the OPLS-DA model was used to cluster the vinegars, where Q^2^ > 0.5 indicated a good fitting degree. The model showed that the values of R^2^X, R^2^Y, and Q^2^ were 0.492, 0.658, and 0.573, respectively, confirming the difference in four types of vinegars (Appendix A). Forty components showed VIP > 1 and *p* < 0.05, and thus were regarded as the differential components (Figure 5, right). The heat map with these components clearly shows the difference between each type of vinegar, with the exception of SAV2, which was confused with ZAV (Figure 5, left). However, the samples from each brand could not be clustered together, suggesting that the technology rather than the manufacturer contributed to the convergence of a type of vinegar. In addition, the samples clustering analysis suggested a similarity between ZAV and FMV, as well as similarity between SAV and SBV. The most significant difference was observed between FMV and SBV, which could be caused by the components belonging to clusters 1, 2, 3, and 5. Cluster 3 correlated positively to FMV but negatively to SBV, and the other three clusters presented opposite correlations.

### 3.6. Marker Screening and the Correlation to Technology

With Lefse, random forest, and PLS-DA, 38, 25, and 23 significantly differential substances were extracted, respectively (Figure 6A–D and Appendix A). In total, 9 substances, including 5 volatiles (2-methylpropyl acetate, 3-methylbutyl acetate, 2,3,5-trimethylpyrazine, 2,3,5,6-tetramethylpyrazine, and 1,2-dihydro-1,1,6-trimethylnaphthalene), 2 amino acids (GABA and Ala), and 2 organic acids (acetic acid and fumaric acid), were identified to be common in these three methods and OPLS-DA (Figure 6E). Interestingly, the nine substances could be used as markers to define the four vinegars (Figure 6F). The results supported the importance of the classification of volatiles on vinegars [11]. Moreover, the markers could be associated with basic technologies. The high contents of 2,3,5,6-tetramethylpyrazine and 2,3,5-trimethylpyrazine in SAV related mostly to the smoking process, though 2,3,5,6-tetramethylpyrazine could be from the metabolisms of *Lactobacillus*, *Acetobacter*, and *Bacillus* [9,44,45]. Rich lactic acid suggested the abundant LAB in SBV (Appendix A). Then, LAB played an important role in GABA production [46], and *Weissella confusa* positively correlated with the GABA and Ala in SBV [6]. C_13_-polyols had been elucidated as the progenitors of 1,2-dihydro-1,1,6-trimethylnaphthalene and showed remarkable content differences depending on the production areas [47]. Thus, the special contribution of 1,2-dihydro-1,1,6-trimethylnaphthalene to SBV should be attributed to local raw materials. The *Monascus* activity would be a reason for 2-methylpropyl acetate detected only in FMV, referring to the generation of 2-methylpropionate during *Monascus* fermentation [48]. Additionally, 2-methylpropanol was reported in *Monascus* wine [49].

For some famous styles of traditional products, the fermentation process is initiated with spontaneous microbial inoculum. In the large-scale fermentation industry, the use of starter cultures has become standard practice to control consistency and process efficiency. Most production methods, such as technologically advanced equipment, are undergoing sustainable development. However, local microflora are still important to achieve stylistic distinction and complexity [50]. Modified technologies can result in a substantial change in the diversity of fermentation organisms. For instance, during the mechanized fermentation of Maotai-flavor *Daqu*, the temperature and acidity changed, which disturbed the growth of the major functional bacteria [51]. The separation of the four types of vinegars should be attributed essentially to the domestication of the environmental microorganisms to be fixed communities, which are essential to the development of flavor [52]. As well, the raw materials providing substrates for microorganisms affected the dynamic structure of microflora [52]. Many other uncontrolled factors also cause the flavor fluctuation. The aging process has the ability to enhance the flavor [25], and a distinct regional environment and basic technology exacerbates the differences in the types of vinegar [13]. Overall, though the detailed diversity of technology cannot be traced for our samples, the results confirmed the important impacts of basic technology on the characteristics of vinegar.

## 4. Conclusions

The flavor characteristics of commercial Chinese ‘four famous vinegar’ were revealed based on the profiles of volatiles, amino acids, and organic acids. The concentrations of all of the chemical compositions changed greatly among the vinegars. Most amino acids and organic acids and a few volatiles were common to all vinegars, but a dozen volatiles were shared by each type of vinegar. Among all compounds, 2-methylpropyl acetate was unique due to its common occurrence in just one type of vinegar (FMV), yet it was undetectable in others, which made it important to cluster the FMV. Heating treatments facilitated similar aroma characteristics of SAV, ZAV, and SBV. The classification of the four vinegars could be achieved with all compositions or nine extracted markers (2,3,5-trimethylpyrazine, 2,3,5,6-tetramethylpyrazine, GABA, Ala, 1,2-dihydro-1,1,6-trimethylnaphthalene, 2-methylpropyl acetate, 3-methylbutyl acetate, acetic acid, and fumaric acid). Further, difference analysis suggested the most significant difference between FMV and SBV. While the correspondence between markers and technologies was rationally demonstrated, the importance of the basic process on the characteristics of ‘four famous vinegars’ should be approved.

## Figures and Tables

**Figure 1 foods-12-01865-f001:**
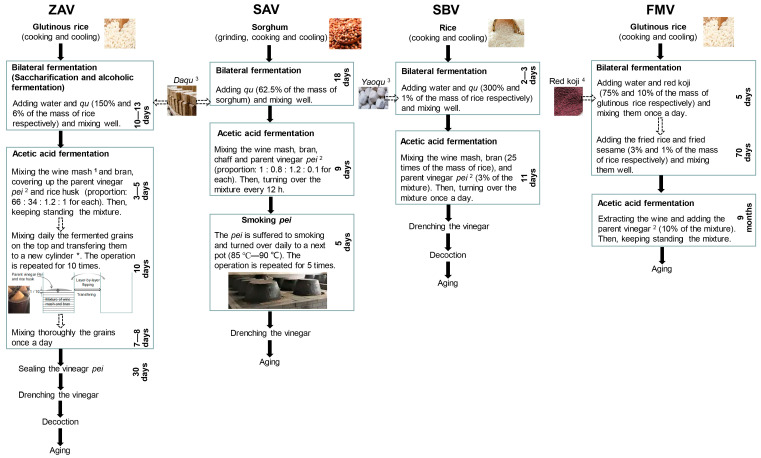
The technological diagrams of ZAV, SAV, SBV, and FMV. ^1^ Wine mash, the mature mixture of alcohol fermentation. ^2^ Parent vinegar (*pei*), the mature mixture taken from the last batch of AAF. ^3^ Daqu and Yaoqu, a saccharifying agent which possess amylase, protease, and a variety of fermentation organisms. ^4^ Red koji, a saccharifying agent rich in *Monascus*. * Rice husk (2.2% of the wine mash) should be added before mixing from the second day. ZAV procedure referred to by Zhao et al. [7]. SAV procedure referred to by Wu et al. [8]. SBV procedure referred to by Zhang et al. [9] and Bao et al. [10]. FMV procedure referred to by Jiang et al. [6] and Bao et al. [10].

**Figure 2 foods-12-01865-f002:**
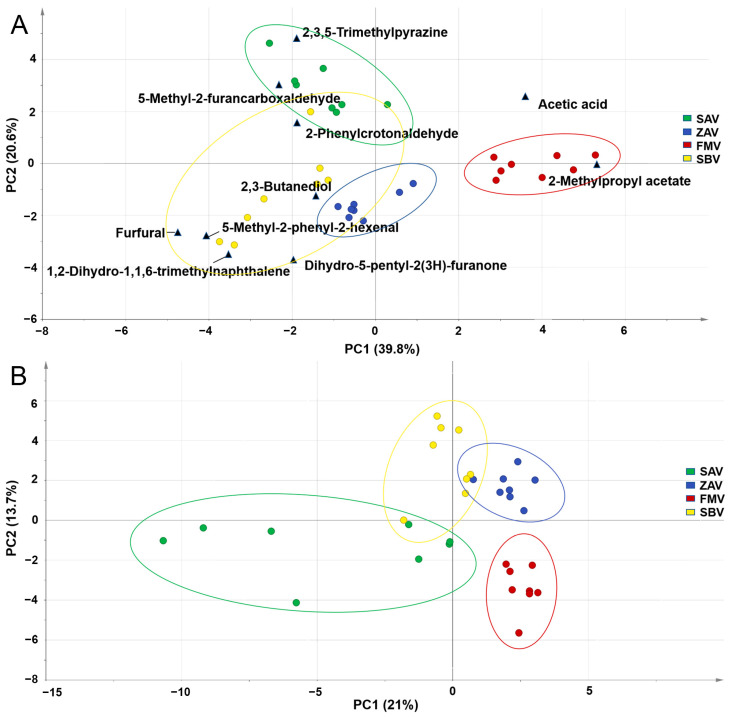
The PCA biplot analysis for SAV, ZAV, FMV, and SBV based on the (**A**) 22 volatiles and (**B**) 68 volatiles. Top 10 contributing compounds are labeled in (**A**).

**Figure 3 foods-12-01865-f003:**
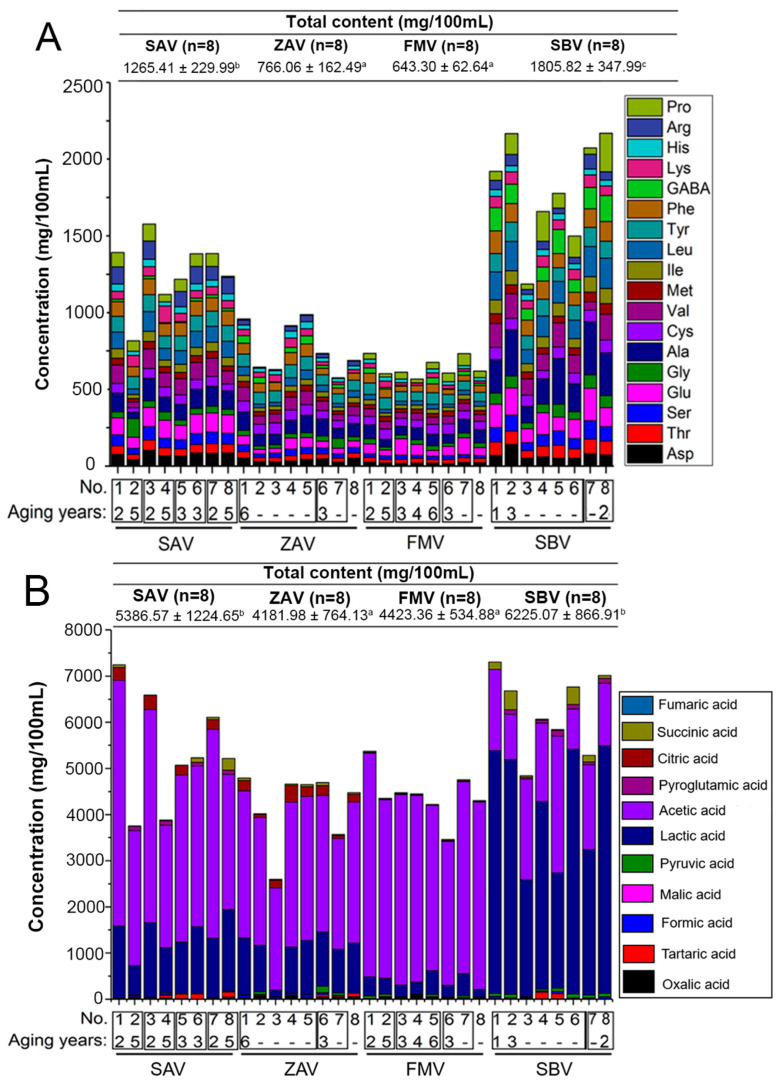
The distribution of (**A**) free amino acids and (**B**) organic acids in SAV, ZAV, FMV, and SBV. Contents of total amino acids and organic acids were depicted on the top. Different letters indicated a significant difference, *p* < 0.05. The sample numbers and aging years were marked at the bottom. The box represented the samples with same brand.

**Figure 4 foods-12-01865-f004:**
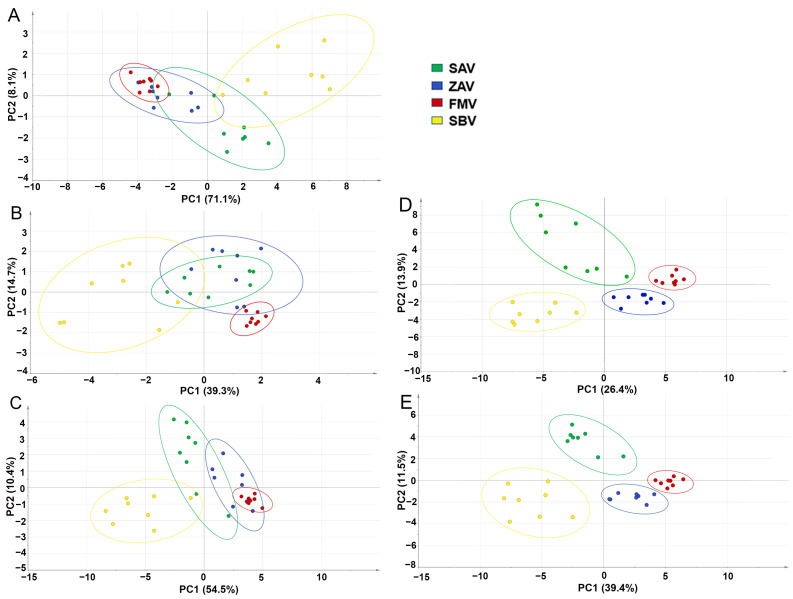
The PCA analysis for SAV, ZAV, FMV, and SBV based on (**A**) amino acids, (**B**) organic acids, (**C**) amino acids, and organic acids; (**D**) all flavors including all 68 volatiles and (**E**) 22 volatiles, amino acids, and organic acids.

**Figure 5 foods-12-01865-f005:**
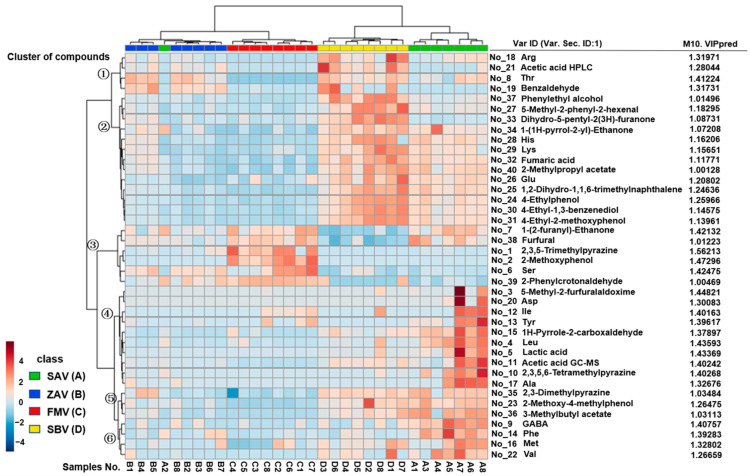
The heat map analysis with 40 differential components (VIP > 1 and *p* < 0.05). OPLS-DA was performed with 68 volatiles, amino acids, and organic acids. The differential components with VIP > 1 and *p* < 0.05 were listed on the right. The numerical order represented the component with decreased VIP value. The number on the left represented the cluster of components. The sample numbers were marked at the bottom.

**Figure 6 foods-12-01865-f006:**
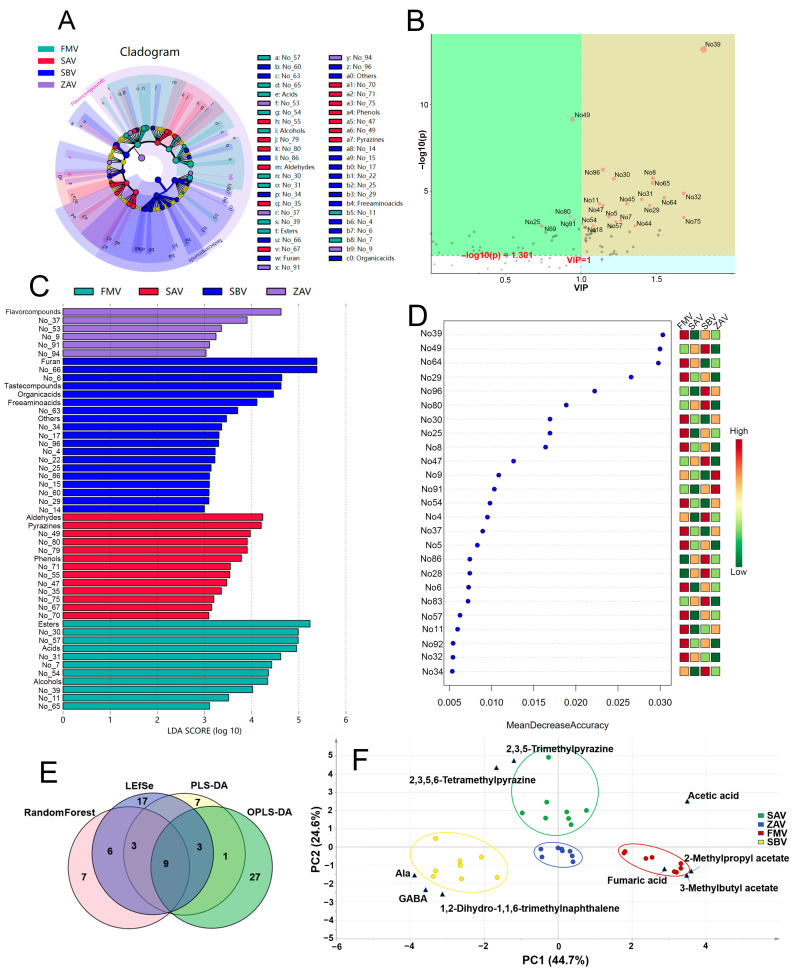
Differential components analysis for SAV, ZAV, FMV, and SBV. (**A**,**B**) Lefse analysis, LDA > 3. (**C**) PLS-DA analysis. (**D**) Random forest analysis. (**E**) Venn diagram. (**F**) PCA biplot performed with the 9 differential substances (2,3,5-trimethylpyrazine, 2,3,5,6-tetramethylpyrazine, GABA, Ala, 1,2-dihydro-1,1,6-trimethylnaphthalene, diethyl butanedioate, acetic acid, and fumaric acid).

## Data Availability

Data is contained within the article and Appendix A.

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
