# Peer review of "Characteristics and Discrimination of the Commercial Chinese Four Famous Vinegars Based on Flavor Compositions"

_foods, 2023, doi:10.3390/foods12091865_

Round 1

Reviewer 1 Report

Review for

 Characteristics and discrimination of the commercial Chinese four famous vinegars based on flavor compositions

 Shanxi aged vinegar (SAV), Zhenjiang aromatic vinegar (ZAV), Sichuan bran vinegar  (SBV) and Fujian monascus vinegar (FMV) are the representative Chinese traditional vinegars

 In  this study, compositions of commercial vinegar were investigated to evaluate the influence of diverse technologies on their distinct flavor.

 ------------------------------------------

 only 5 volatiles were detected in all vinegars whereas a dozen volatiles were common to  each type of vinegar

 it appears to be a low number of volatiles, compared to vinegars produced in other parts of the world

 Vinegars are natural products manufactured by two-step fermentation. One of the most famous is balsamic vinegar, especially that produced in Modena, Italy. Its unique production process positively distinguishes it from other vinegars. There are basically three types of balsamic vinegar: common balsamic vinegarBalsamic vinegar of Modena, and Traditional balsamic vinegar of Modena. The chemical analysis of these vinegars is mainly carried out by using gas or liquid chromatography, often coupled to mass spectrometric detection. Although gas chromatography is generally used for the determination of the overall profile of volatile organic compounds, furfurals, phenolic compounds, and organic acids, high-performance liquid chromatography is typically applied for the determination of amino acids, sugars, and polyphenols. The two complementary techniques, the combination of which is useful for the detailed characterization of balsamic vinegars, are reviewed and discussed in this article.

10.3390/app12188946

--------------------------------------------------------------------

 Why comparing only Chinese vinegars? Why vinegars from other parts of the world were not introduced in the study?

 ------------------------

 Shanxi aged vinegar (SAV), Zhenjiang aromatic vinegar (ZAV), Sichuan bran vinegar  (SBV) and Fujian monascus vinegar (FMV)

Adulteration? Frauds? In the vinegar trade

-----------------------------

  Further, the correlation of identification markers to vinegars stressed the contributions of  smoking process, raw materials and Monascus inoculum to SAV, SBV and FMV clustering respectively.

 When Monascus is used, no mycotoxin citrinin detected?

 --------------------------

  Unlike the European vinegar which is mainly produced from fruits, Chinese traditional vinegar is  made mainly from grains.

 Other parts in the world where vinegars from grains are manufactured?  Or limited to Asia?

 ------------------------------

   Author Response

Response to Reviewer 1 Comments

 Point 1: It appears to be a low number of volatiles, compared to vinegars produced in other parts of the world.

Response 1: Thank you. In our study, SPEM-GC-MS was used for determination of the profile of volatiles. Dozens of volatiles were identified from SAV (57), ZAV (46), FMV (45) and SBV (44). And, 20-30 volatiles were identified in samples, which were similar with other studies of Chinese traditional vinegars. For example, 23 and 19 volatiles were identified in SAV [1] and FMV [2] respectively.

  We have read the article you recommended and other articles. About 50 aroma compounds were present in balsamic vinegar samples [3]. The materials and whole techonical process including the aging process should be the main factors resulting in a low number of volatiles in our samples compared to balsamic vinegar [3-4]. Extraction by organic solvent such as diethyl ether would detect different profiles of compositions [3].

  1. Zhu H, Zhu J, Wang L, et al. Development of a SPME-GC-MS method for the determination of volatile compounds in Shanxi aged vinegar and its analytical characterization by aroma wheel. Journal of food science and technology, 2016, 53: 171-183.
  2. Liu L, Hu H, Yu Y, et al. Characterization and identification of different Chinese fermented vinegars based on their volatile components. Journal of Food Biochemistry, 2021, 45(3): e13670.
  3. Farhana R. Pinu, Samuel De Carvalho-Silva, Ana Paula Trovatti Uetanabaro and Silas G. Villas-Boas. Vinegar Metabolomics: An explorative study of commercial balsamic vinegars using gas chromatography-mass spectrometry. Metabolites, 2016, 6(3), 22.
  4. Kašpar M, ÄŒesla P. Characterization of Balsamic Vinegars Using High-Performance Liquid Chromatography and Gas Chromatography. Applied Sciences, 2022, 12(18): 8946.

We added some discuss in the revised paper, as is ‘In samples, 20-30 volatiles were identified (Table S2), which were similar with other studies of Chinese traditional vinegars (Zhu, et al., 2016; Liu, et al., 2021) but lower compared to the balsamic vinegar (Farhana, et al., 2016; Kašpar, et al., 2022). The materials and whole techonical process including the aging process should be the main factors resulting in the difference. Combinational use of volatile extraction by organic solvent would expand the overall profile of volatile (Farhana, e al., 2016).’ in line 224-229.

We hope the correction would saitisfiy you.

 Point 2: Why comparing only Chinese vinegars? Why vinegars from other parts of the world were not introduced in the study?

Response 2: Thank you for your suggestion. There are many other traditional vinegars in China, but Shanxi aged vinegar (SAV), Zhenjiang aromatic vinegar (ZAV), Sichuan Bran vinegar (SBV), and Fujian monascus vinegar (FMV) as the top representative vinegars have large market share in China. Consequently, we think it is important to discriminate the four famous vinegars and evaluate the influence of complicated industrial technologies on their characteristics. The study may promote the basis for defining properties and developing quality standards for the ‘four famous vinegars’.

We used the commercially available vinegars as the samples, but limit samples of a type of vinegar from other parts of the world could be purchased in local market. This was also one reason why vinegars from other parts of the world were not included. However, we believe that the suggestion is a good idea.

Point 3: Shanxi aged vinegar (SAV), Zhenjiang aromatic vinegar (ZAV), Sichuan bran vinegar  (SBV) and Fujian monascus vinegar (FMV). Adulteration? Frauds? In the vinegar trade.

Response 3: Thank you. In the Chinese traditional vinegar trade, many measures are used to ensure food quality, such as Chinese National Standard GB/T 18187-2000 and GB 2719-2018, industry standard and company standard. Only when an enterprise has production qualifications can production license be obtained. All the four vinegars are certified as Protected Geographic Indication in China. Commercial adulteration may exist. But, to our known, there were no reports about adulteration for the four vinegars.

Point 4: When Monascus is used, no mycotoxin citrinin detected?

Response 4: Thank you for your suggestion. Elimination of citrinin is essential in Monascus-related food products. Study has shown that citrinin concentrations decreased considerably in Monascus vinegar fermentation and is much lower than the maximum allowed level, even when citrinin was artificially introduced [1]. That study suggested eliminating citrinin by vinegar fermentation is of merit from the viewpoint of food safety. And, in Chinese National Standards including GB 1886.19-2015 (Red koji rice), QB/T 5188-2017 (Red koji for brewing), GB/T 1886.181-2016 (Food Additive Monascus Red) and so on, maximum allowable content of mycotoxins in products is executed. Mearsures such as strain selection were also applied to control the mycotoxins in monascus products. For example, Monascus pilosus MS-1 is selected as a high-monacolin K-producing but citrinin-free strain [2].

  1. Hsieh C W, Lu Y R, Lin S M, et al. Stability of monacolin K and citrinin and biochemical characterization of red-koji vinegar during fermentation. Journal of agricultural and food chemistry, 2013, 61(30): 7276-7283.
  2. Feng Y, Shao Y, Zhou Y, et al. Effects of glycerol on pigments and monacolin K production by the high-monacolin K-producing but citrinin-free strain, Monascus pilosus MS-1. Eur Food Res Technol, 2014, 240: 635–643.

 Point 5: Other parts in the world where vinegars from grains are manufactured?  Or limited to Asia?

Response 5: Thank you for your suggestion. The grain vinegar are manufactured mainly in East Asia such as China, Japan, and Korea. We have added this and rewrote the sentence in the new paper to make the discription clear, in line 42-43.

Reviewer 2 Report

The work is interesting, written understandably despite advanced analyzes and statistical elaboration.

Although the chemical diversity of different types of vinegar is rather obvious, but finding markers characteristic of these products can be useful in food analysis

In Table 1, the SD is larger than average value. Is it possible with this type of data? This should be explained when discussing the results, because from mathematical point of view it is incorrect. With such SD values, two different adjoin peaks can sum up and the interpretation may be wrong.  This need to be explained or corrected

Author Response

Response to Reviewer 2 Comments

 Point 1: In Table 1, the SD is larger than average value. Is it possible with this type of data? This should be explained when discussing the results, because from mathematical point of view it is incorrect. With such SD values, two different adjoin peaks can sum up and the interpretation may be wrong.  This need to be explained or corrected.

Response 1: Thank you. In our study, we selected eight samples for each type of vinegar, and each sample was detected in triplicate. Reproducibility is fine for a sample. For example, the content of 2,3,5,6-Tetramethylpyrazine in SAV1 are 850, 864.7 and 848.2, that is presented as 848.9 ± 13.29. But, when the statistic was performed for several samples, the SD value appears large (886.20 ± 936.10) with whether Tukey analyse or Duncan analyse. Similar phenomenon occurred in other study [1]. We suggested that it is caused by the great content difference among samples. We have explained it in the revised paper, line 242-244. We hope the correction would saitisfiy you.

  1. Feng Y, Su G, Zhao H, et al. Characterisation of aroma profiles of commercial soy sauce by odour activity value and omission test. Food Chemistry, 2015, 167: 220-228.

Reviewer 3 Report

The paper is interesting and innovative. The proposed methodology can also be used in the authenticity analysis of other types of fermented food products. Overall, the publication deserves to find a place in the journal. I will propose a major revision rather for purely technical reasons, as some of the notes should find a place in the final version of the publication.

Please improve the quality of figures 1, 3, 5, 6 and others. At the moment, the texts in them are very small and do not make it possible to read them, which puts at risk the giving of qualitative conclusions.

Some of the methods of statistical analysis need to be explained in more detail, as they are not used very often - for example, those on which Fig. 5 and 6 were prepared.

In my opinion, Table 1 should be as an appendix, and the figures reflecting the PCA should be in the main text of the publication. This will highlight the fact that the different types of vinegar are distributed into groups based on the main components defined by the analysis.

Author Response

Response to Reviewer 3 Comments

 Point 1: Please improve the quality of figures 1, 3, 5, 6 and others. At the moment, the texts in them are very small and do not make it possible to read them, which puts at risk the giving of qualitative conclusions.

Response 1: Thank you for your suggestion. We have already updated the new figures in our manuscript. The texts in figure 6A-D appear still small, but can not be adjusted to be larger during the images generation by software.

Point 2: Some of the methods of statistical analysis need to be explained in more detail, as they are not used very often - for example, those on which Fig. 5 and 6 were prepared.

Response 2: Thank you for your suggestion. We have added the explaination in section 2.9 of Materials and methods, line 208-217.

‘Orthogonal partial least squares discriminant analysis (OPLS-DA) can be used for fea-ture extraction and classification of components and can identify variables related to different classifications determining the contribution of variables to sample differences (Santos, et al., 2018). LDA effect size (LEfSe) can describe the differences between groups, and screen variables with the most significant differences between them (Moreno, et al., 2020). Random Forest is a machine learning algorithm based on decision trees, which can perform feature selection from a large number of variables (Wu, et al., 2009). Partial least squares discrimination analysis (PLS-DA) is a multivariate statistical analysis based on multiple linear regression. It is also used to deal with the relationship between multiple independent or dependent variables (SzymaÅ„ska, et al., 2012). ’

Point 3: In my opinion, Table 1 should be as an appendix, and the figures reflecting the PCA should be in the main text of the publication. This will highlight the fact that the different types of vinegar are distributed into groups based on the main components defined by the analysis.

Response 3: Thank you for your suggestion. We have arranged the figures and tables in the manuscript according to you. We hope the correction would saitisfiy you.

Reviewer 4 Report

The manuscript is aimed at researching characteristics and discrimination of the commercial Chinese four famous vinegars based on flavor compositions.

The studies were performed using modern methods. The results obtained are discussed with the works of other authors. References used appropriately.

However, the purpose of the research is not clearly formulated. In this regard, it is not clear what significance the obtained results have and how they can be used in the future.

Author Response

Response to Reviewer 4 Comments

 Point 1: The purpose of the research is not clearly formulated. In this regard, it is not clear what significance the obtained results have and how they can be used in the future.

Response 1: Thank you for your valuable feedback to improve the quality of our manuscript. In our study, classification of the four vinegars could be achieved well, and several markers could be used to demonstrat the importance of basic process on the characteristics of vinegars. We think it could contribute to define the properties and develope the quality standards for the vinegars. For instance, aboundant 2-methylpropyl acetate or its derivatives could be typical marker to define Fujian monascus vinegar (FMV) due to the Monascus activity. We have added the purpose of the study and the potential application in the Abstract. We hope the correction would saitisfiy you.

Reviewer 5 Report

The article “Characteristics and discrimination of the commercial Chinese four famous vinegars based on flavor compositions” aims to evaluate the influence of diverse technologies on the composition of commercial vinegar and their distinct flavor. For that to be achieved, the profiles of the volatiles, free amino acids, and organic acids, were investigated in four commercial vinegars.

When retrieving similar works on the internet, I found the following:

 Hu, Yong and Zheng, Chuanyang and Chen, Haiyin and Wang, Chao and Ren, Xiyue and Fu, Shiming and Xu, Ning and Li, Panheng and Song, Jinyi and Wang, Chao, Characteristics and Discrimination of the Commercial Chinese Four Famous Vinegars Based on Flavor Compositions. Available at SSRN: https://ssrn.com/abstract=4274035 or http://dx.doi.org/10.2139/ssrn.4274035

 The information and data are very similar to what is presented in the current manuscript…So, I do not think that duplication of data is the most correct and ethical form to publish scientific work.

Author Response

Response to Reviewer 5 Comments

 Point 1: When retrieving similar works on the internet, I found the following:

Hu, Yong and Zheng, Chuanyang and Chen, Haiyin and Wang, Chao and Ren, Xiyue and Fu, Shiming and Xu, Ning and Li, Panheng and Song, Jinyi and Wang, Chao, Characteristics and Discrimination of the Commercial Chinese Four Famous Vinegars Based on Flavor Compositions. Available at SSRN: https://ssrn.com/abstract=4274035 or http://dx.doi.org/10.2139/ssrn.4274035

The information and data are very similar to what is presented in the current manuscript…So, I do not think that duplication of data is the most correct and ethical form to publish scientific work.

Response 1: Thanks. We feel great sorry for the mistake. The similar works you found on the internet are preprints. We do not know how the preprint be created, but it is certainly not a published paper. We have already removed it from SSRN’s eLibrary. We ensure that there is no multiple submissions for the paper.

Round 2

Reviewer 3 Report

The authors has took into account all my comments.

Reviewer 4 Report

Dear Authors,

You fully answered the comments of the reviewers and made the necessary corrections to the manuscript.

Reviewer 5 Report

If the similar work  found on the internet are preprints and they were removed from the  SSRN’s eLibrary i think the article may be published.

It is an excellent work.